# Influence of cardiovascular risk factors and treatment exposure on cardiovascular event incidence: Assessment using machine learning algorithms

**Sara Castel-Feced**[1,2,3]*, **Sara Malo**[1,2,3], **Isabel Aguilar-Palacio**[1,2,3], **Cristina Feja-Solana**[2,3,4], **José Antonio Casasnovas**[5,6], **Lina Maldonado**[2,3,7‡], **María José Rabanaque-Hernández**[1,2,3‡]

1 Microbiology, Pediatrics, Radiology, and Public Health, University of Zaragoza, Zaragoza, Spain, 2 Fundación Instituto de Investigación Sanitaria de Aragón (IIS Aragón), Zaragoza, Spain, 3 GRISSA Research Group, Zaragoza, Spain, 4 Directorate of Public Health, Government of Aragon, Zaragoza, Spain, 5 Hospital Universitario Miguel Servet, Instituto de Investigación Sanitaria Aragón (IIS Aragón), CIBERCV, Zaragoza, Spain, 6 Department of Medicine, Psychiatry and Dermatology, University of Zaragoza, Zaragoza, Spain, 7 Department of Applied Economic, University of Zaragoza, Zaragoza, Spain

‡ LM and MJRH also contributed equally to this work and served as senior co-authors.
* scastelf@unizar.es

**Data Availability Statement:** Data were provided by the Aragon Health Sciences Institute (IACS), Spain, so authors do not have permission to share

## Abstract

Assessment of the influence of cardiovascular risk factors (CVRF) on cardiovascular event (CVE) using machine learning algorithms offers some advantages over preexisting scoring systems, and better enables personalized medicine approaches to cardiovascular prevention. Using data from four different sources, we evaluated the outcomes of three machine learning algorithms for CVE prediction using different combinations of predictive variables and analysed the influence of different CVRF-related variables on CVE prediction when included in these algorithms. A cohort study based on a male cohort of workers applying populational data was conducted. The population of the study consisted of 3746 males. For descriptive analyses, mean and standard deviation were used for quantitative variables, and percentages for categorical ones. Machine learning algorithms used were XGBoost, Random Forest and Naïve Bayes (NB). They were applied to two groups of variables: i) age, physical status, Hypercholesterolemia (HC), Hypertension, and Diabetes Mellitus (DM) and ii) these variables plus treatment exposure, based on the adherence to the treatment for DM, hypertension and HC. All methods point out to the age as the most influential variable in the incidence of a CVE. When considering treatment exposure, it was more influential than any other CVRF, which changed its influence depending on the model and algorithm applied. According to the performance of the algorithms, the most accurate was Random Forest when treatment exposure was considered (F1 score 0.84), followed by XGBoost. Adherence to treatment showed to be an important variable in the risk of having a CVE. These algorithms could be applied to create models for every population, and they can be used in primary care to manage interventions personalized for every subject.

the data. The permission obtained implies the exclusive use of the data by researchers who authored the present study. Thus, this information cannot be published or shared with other institutions. Data access requests should be addressed to the IACS through https://www.iacs.es/. Source code is openly available at https://github.com/saracf623/machinelearning.

**Funding:** This study was supported by Proyecto del Fondo de Investigación Sanitaria, Instituto de Salud Carlos III (Ministerio de Ciencia e Innovación) and the European Fund for Regional Development (FEDER) (PI17/01704) and by the Grupo de Investigacion en Servicios Sanitarios de Aragón (GRISSA) [B09-23R] of the IIS Aragon, funded by the regional Government of Aragon, Spain. It was also partly supported to SCF by Gobierno de Aragón with a grant for postgraduate research contracts (IIU/796/2019). The funders had no role in study design, data collection and analysis, decision to publish, or preparation of the manuscript.

**Competing interests:** The authors have declared that no competing interests exist.

## Introduction

Cardiovascular diseases (CVD) are the leading cause of morbidity and mortality worldwide and are responsible for 32% of all global deaths [1]. CVD prevention guidelines emphasize the importance of primary CVD prevention, applying lifestyle changes and medicating according to the individual's overall CV risk, which reflects the contributions of multiple CV risk factors [2]. Several scoring systems are used to predict CVD risk, with the ultimate goal of establishing preventive interventions, both pharmacological and non-pharmacological. These scoring systems, which include the Framingham Risk Score and the Systematic Coronary Risk Evaluation (SCORE), have been developed for specific populations, without considering whether subjects are being treated for any cardiovascular risk factor (CVRF), and also suffer from certain methodological limitations [3–5] due to correlation between variables, non-linearity of variables, and the possibility of over-fitting. On the other hand, the knowledge acquired from an ever-growing body of medical data generated in daily clinical practice is providing researchers with valuable insights into medical conditions [6, 7].

Machine learning techniques have been widely applied [6, 8] to probe these huge datasets and overcome some of the aforementioned limitations of scoring systems. Machine learning can be used to generate models that better predict risk, thereby increasing the efficiency, objectivity, and reliability of the diagnostic process [3, 4, 6, 9]. Specifically, these supervised learning techniques use existing data to train models by learning patterns that will be later applied to predict another variable. When applying these techniques to disease research, there are some problems which have to be considered during the analysis. These problems are related to the interpretability of the models and to data imbalance, quality and quantity. Poor interpretability is due to the fact that they work like black boxes, making it difficult to interpret their results. This problem has been overcome by the development of different methods, implemented in different R libraries, that try to set the importance of each feature in the prediction [10–12]. Imbalance data is because in health research always there are fewer people who are sick than those who are healthy, so usually data are imbalance. Finally, quality and quantity of medical data usually are low because of the sources of information available and because of accessibility and ethical issues [13].

There are different kinds of supervised machine learning. When the variable to be predicted is categorical (e.g. a cardiovascular event), classification machine learning techniques [3, 4] such as Naïve Bayes (NB) algorithms and ensemble methods are used [12, 14, 15]. Ensemble methods include bagging and boosting methods such are Random Forest (RF), XG Boost [12, 14, 15].

Machine learning techniques represent a promising approach for CVD risk prediction [3, 4, 12, 14–16] and the advantages they offer over existing scoring systems. In the present study, we analyzed the ability of three machine learning algorithms, NB, RF and XG Boost, to predict the appearance of cardiovascular events (CVE) and analysed the influence of different CVRF-related variables on CVE.

## Methodology

### Study design and participants

This longitudinal cohort study was conducted within the framework of the Aragon Workers' Health Study (AWHS), a prospective, longitudinal cohort study of workers at an automobile assembly plant located in Figueruelas (Zaragoza, Spain). Recruitment began in February 2009 and ended in December 2010. Since then, data have been collected from the participant's annual medical tests and their utilization of health services has been monitored. Further information on the AWHS can be found in Casasnovas et al. [17].

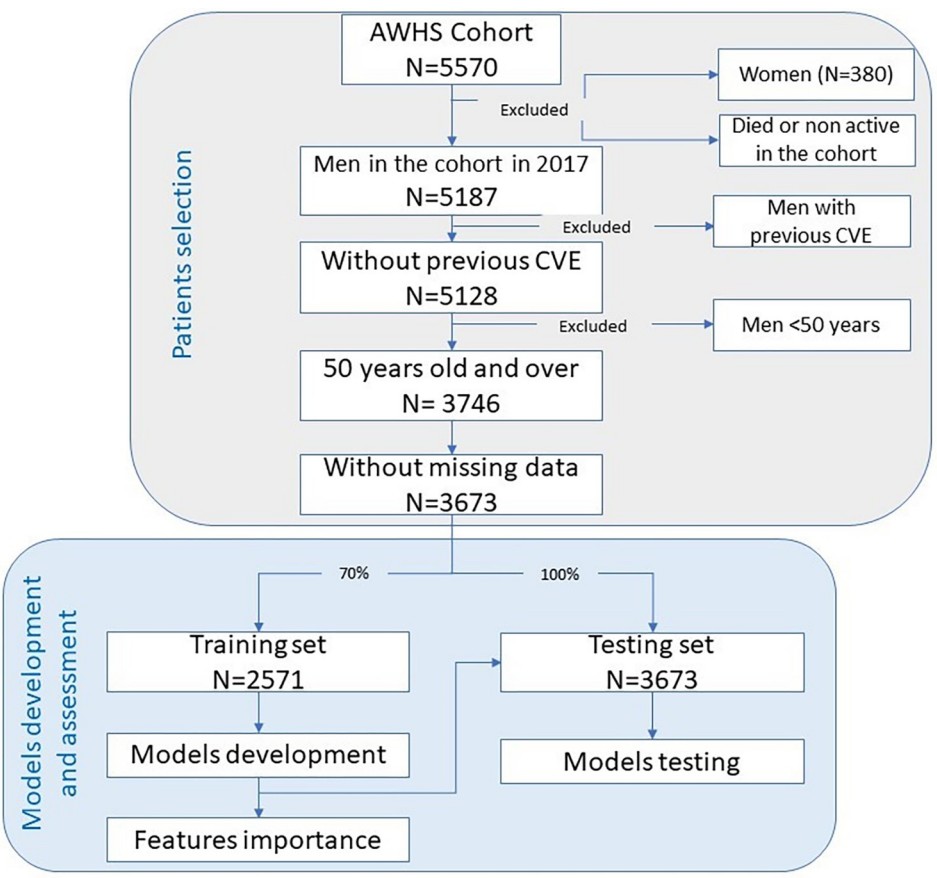

**Fig 1. Flowchart depicting the study population and model development.**

All subjects included in our analysis were male (N = 3,746), aged ≥50 years, with no previous medical history of CVE. Females were excluded owing to the low number in the cohort (N = 380) and individuals aged <50 years were excluded owing to the low incidence of CVE in this age group. After selecting the study population, we identified individuals who experienced a CVE at any moment between inclusion in the cohort and December 31, 2019, and recorded the date and nature of each CVE. The selection of patients is explained in Fig 1.

## Data source and variables

Data from several sources were used: the AWHS cohort; BIGAN; and the General Direction of Public Health. BIGAN [18] is a health data hub that gathers data from the Aragon Public Health Service. These data are available for research upon request. From the AWHS study we included data from workers' annual medical tests (including blood tests). From BIGAN we obtained data from (i) the Pharmaceutical Dispensation Database, which collects information on the dispensing date, Anatomical Therapeutic Chemical (ATC) code, number of defined daily doses (DDD), and the number of packages dispensed by pharmacies and funded by the Aragon Health Service; (ii) the MBDS (Minimum Basic Data Set), which registers diagnoses and dates of hospitalizations; and (iii) the Emergency database which records diagnoses and dates pertaining to emergency service utilization. Finally, information on the date and cause of death was obtained from the Aragon Mortality Registry via the General Direction of Public Health.

**Table 1. Variables sources and missing data.**

| Variables | AWHS | BIGAN | | | General Direction of Public Health | |
| --- | --- | --- | --- | --- | --- | --- |
| | Annual medical test | Pharmaceutical dispensation database | Minimum Basic Data Set | Emergency Database | Aragon Mortality Registry | NA |
| **Hypertension** | X | X | | | | 26 (0.5%) |
| **Hypercholesterolemia** | X | X | | | | 14 (0.3%) |
| **Diabetes** | X | X | | | | 27 (0.6%) |
| **Physical status** | X | | | | | 57 (1.2%) |
| **Age** | X | | | | | 0 |
| **Treatment exposure** | | X | | | | 0 |
| **CVE** | | | X | X | X | 0 |

The source of the variables and number of missing data is summarised in Table 1.

**Explanatory variables.** *CVRF definition.* The following CVRFs were considered: hypertension, hypercholesterolemia (HC), diabetes mellitus (DM), and physical status, as calculated based on body mass index (BMI). Smoking status was not considered as corresponding data were not available for the period analysed. CVRFs were identified for the year preceding the first CVE in individuals with a CVE, and for 2019 for individuals with no CVE during the study period.

Medical and blood test findings and data from the Pharmaceutical Dispensation Database were examined to identify CVRFs. Subjects were classified as suffering from the CVRF if they were registered in at least one of those databases as such. CVRFs were identified from medical and blood test data applying the following cut-off points, as recommended by European CVD prevention guidelines [2]: overweight was defined as a BMI $\geq$25 and <30, and obesity as a BMI $\geq$30; hypertension as diastolic blood pressure $\geq$90 mmHg and/or systolic blood pressure $\geq$140 mmHg; HC as total cholesterol $\geq$200 mg/dl or LDL-cholesterol $\geq$115 mg/dl; and DM as fasting serum glucose $\geq$126 mg/d.

Based on data from the Pharmaceutical Dispensation Database, individuals were considered to have hypertension if they had filled at least one prescription corresponding to the following ATC codes: C02 (antihypertensives), C03 (diuretics), C07 (beta-blocking agents), C08 (calcium channel blockers), and C09 (agents acting on the renin–angiotensin system). Since diuretics and beta-blocking agents are also prescribed for other indications, dispensation of these drugs was only considered an indicator of hypertension if the individual filled at least three distinct dispensations within the same year [19]. Participants were considered to have HC if they filled at least one dispensation corresponding to ATC code C10 (lipid modifying agents) and DM if they filled least one dispensation corresponding to ATC code A10 (drugs used in diabetes).

*Treatment exposure.* *Treatment exposure* was determined by quantifying adherence to the treatment.

To standardise terminology related to adherence to pharmacological therapies, the European Ascertaining Barriers for Compliance (ABC) project proposed a Taxonomy of Adherence [20] consisting of three components: initiation, implementation, and discontinuation. Treatment *initiation* corresponds to intake of the first dose of a prescribed medication. The process continues with *implementation* of the dosing regimen, defined as the extent to which a patient's actual dosing corresponds to the prescribed dosing regimen, from treatment initiation until consumption of the last dose. *Discontinuation* indicates the end of therapy, when the next dose to be taken is omitted and no more doses are taken thereafter.

In the present study, our analysis focused on the implementation phase. Adherence to HC, hypertension, and DM treatments was determined separately for each participant and represented as the Proportion of Days Covered (PDC), calculated as a percentage. PDC is an index calculated as the number of days covered by the medicines dispensed by the pharmacy divided by the number of days that the subject should have had covered. In this study, the denominator for PDC was 365 days, except in cases in which subjects started treatment once the follow-up period had already started. In these cases, the denominator was the number of days from initiation of treatment to the end of the follow-up period. The number of days covered are calculated based on the DDD dispensed to each subject. However, a previous study of our group [21] showed that use of a surrogate value for the daily dose of each drug, calculated based on the usual dosage and form of presentation, provided more accurate results. Therefore, in the present study surrogate values for daily doses were used.

For each subject, the PDC obtained for the three CVRFs was summarized in a new variable: treatment exposure. This variable was classified into 3 possible categories: *fully exposed*, participants who filled prescriptions for the treatment of all identified CVRFs and had a PDC ≥80%; *non-exposed*, participants who filled prescriptions for none of the identified CVRFs or had a PDC <80% for all treatments taken; *partially exposed*, participants who did not fill prescriptions for at least one identified CVRF and had a PDC ≥80% for others or a PDC <80% for some treatment and ≥80% for others.

**Evaluated outcome.** The primary outcome in the current study was the incidence of a CVE during the study period. This CVE was identified based on data from the MBDS, Emergency database, and the Aragon Mortality Registry as follows:

i. For subjects with records in either the MBDS or the Emergency database, data from the former was selected.

ii. For subjects with records in both the MBDS and Emergency databases, checks were performed to determine whether the first record in each database matched in terms of time and diagnosis. If so, data from the MBDS database were chosen. If not, and the MBDS entry predated that in the Emergency database, the former was selected. Conversely, if the record in the Emergency database predated that in the MBDS database, a check was performed to determine whether record corresponded to a non-CVE record in the MBDS database: if so, the record in the Emergency database was rejected; if not, the record in the Emergency database was selected.

iii. Finally, the Aragon Mortality Registry was analysed to identify subjects with a fatal first CVE.

Diagnoses were recorded according to the International Classification of Diseases, 9th revision (ICD-9) in the Emergency database, and according to the International Classification of Diseases, 10th revision (ICD-10) in the MBDS and Aragon Mortality Registry. The following ICD-9 and ICD-10 codes were considered: ICD-9 410–415 and ICD-10 I20-I25 (heart disease); ICD-9 415–417 and ICD-10 I26-I28 (pulmonary heart disease and diseases of pulmonary circulation); ICD-9 427.4, 427.5, 428, 429.2 and ICD-10 I46, I49.0, I50 (other heart diseases); ICD-9 430–438 and ICD-10 G45-G46 and I60-I69 (cerebrovascular diseases); ICD-9 440–445 and ICD-10 I70-I79 (diseases of arteries, arterioles, and capillaries).

## Statistical analyses

For the initial description of the variables included in the study, continuous variables were expressed as the mean and standard deviation and categorical variables as percentages.

**Machine learning models development.** Supervised machine learning algorithms were used to determine the utility of different variables to predict CVE. The process is depicted in Fig 1. These algorithms included RF, XGBoost, and NB. Due to the low number of subjects with missing data, subjects with missing data in any variable were excluded (N = 89). 70% of the data were randomly split to form the training group. The entire sample was used to test the different algorithms, as the sample size of the testing group was small to validate the models obtained. To fit the hyperparameters and avoid over-fitting, the prediction accuracy of all models was tested using 5 and 10 fold stratified cross validation (cv) to estimate F1-score. Results were similar when applying 5 and 10 fold cv, so results shown in the article corresponds to the 5-fold cv.

The following parameters were adjusted for RF models: number of trees, number of features to consider at any given split, maximum depth, which is the maximum number of partitions in the longest branch of the tree, and minimum number of observations in one node. For XGBoost, the parameters adjusted were the number of features to consider at any given split, maximum depth, and the minimum number of observations in one node. These parameters are shown in Table 2. For the NB method, the *a priori* probabilities were 0.9 for non-occurrence of CVE and 0.1 for occurrence of CVE.

These parameters were fitted for each algorithm before its implementation to avoid over-fitting.

Each method was applied twice: first including just CVRFs as variables, and again including both CVRFs and treatment exposure.

**Machine learning models assessment.** Because of the highly imbalanced data that we had, to validate models threshold were moved and selected based on max of f1 score in P-R curves, being 0.1 in all models. To measure the validity of the models, the following measures were taken into account: accuracy, sensitivity, specificity, positive predictive value (PPV) and negative predictive value (NPV). Next, three different tests were applied to evaluate the performance of the model: area under the precision-recall curve (AUC-PR), Log Loss, and F1 score. These scores were selected because different studies recommend them for imbalanced data [22–24].

**Variables importance.** After achieving valid and accurate models, the contribution of each variable to the prediction was extracted using caret R package. Methods applied for each algorithm were different and they give scores in different ranges so, to facilitate comparability, scores obtained for each method were normalized to a scale of 0–1. For RF models, the method applied to compute the contribution of each variable consisted of recoding for each tree the prediction accuracy on the out-of-bag portion. Then each predictor variable was permuted and the same was done. Finally, for all trees, the difference between both accuracies was averaged and normalized by the standard score [10, 11].

**Table 2. Random Forest and XGBoost parameters adjusted.**

|  | CVRF | CVRF AND TREATMENT EXPOSURE |
|---|---|---|
| **RANDOM FOREST** | Num. trees: 5000<br>Number of predictors: 5<br>Max. depth: 3<br>Min. node. size: 10 | Num. trees: 5000<br>Number of predictors: 6<br>Max. depth: 3<br>Min. node. size:10 |
| **XGBOOST** | Number of predictors: 2<br>Max. depth: 6<br>Min. node. size: 38 | Number of predictors: 1<br>Max. depth: 9<br>Min. node. size: 3 |

CVRF, cardiovascular risk factors; Num, number; Max, maximum; Min, minimum.

For XG Boost models, the reduction in the loss function attributable to each variable in sum of squared error in predicting the gradient on each iteration was calculated. Finally, the improvement score for each predictor was averaged across all the trees in the ensemble [11, 25].

Finally, for models developed applying NB a ROC curve analysis was conducted on each predictor. Different cutoffs were applied to the predictor data to predict the class. Then, sensitivity and specificity were computed for each cutoff and the area under ROC curve was calculated using the trapezoidal rule. This area was used as the measure of variable importance [11].

All statistical analyses were performed in the R statistical computing environment (version 4.0.5 Foundation for Statistical Computing, Vienna, Austria).

## Ethical issues

All participants in the AWHS provided prior written informed consent and all collected data were anonymized according with the Spanish Organic Law 3/2018 and the Declaration of Helsinki. The present study was approved by the Clinical Research Ethics Committee of Aragon (project identification code PI17/00042).

## Results

In total, this study included 3,746 participants (mean age, 61.6 years), all of whom were male. The prevalence of hypertension was 66.1%, HC 81.0%, and DM 17.0. The percentage of participants receiving treatment for these conditions was 74.3% for hypertension, 52.7% for HC, and 83.5% for DM. Overweight was recorded in 54.4% of participants and obesity in 30%. The number of CVRFs present was as follows: 1 CVRF, 46.9%; 2 CVRFs, 41.9%; 3 CVRFs, 11.2%. The number of CVEs recorded between January 2010 and December 2019 was 298 (7.9%). Assessment of adherence by pharmacological group indicated a PDC $\geq$80% in 63.3% of participants taking antidiabetics, 78.1% of those taking antihypertensives, and 64.4% of those taking medication for HC.

Mean age, CVRF prevalence, and treatment in subjects with and without a CVE are shown in Table 3. Compared with the non-CVE group, the CVE groups had a higher mean age (1.4 years higher) and a greater prevalence of hypertension, HC, diabetes and obesity. Conversely, the prevalence of overweight was slightly higher in the non-CVE group. Among those with no

**Table 3. Descriptive variables stratified according to incidence of cardiovascular events.**

|  | NO CVE | CVE | P-VALUE |
|---|---|---|---|
| **N** | 3448 (92.05) | 298 (7.95) | |
| **AGE*** | 61.50 (4.82) | 62.90 (4.20) | <0.001 |
| **HYPERTENSION** | 2252 (65.60) | 213 (71.70) | 0.039 |
| **HC** | 2765 (80.30) | 264 (89.2) | <0.001 |
| **DIABETES** | 567 (16.50) | 65 (22.00) | 0.019 |
| **PHYSICAL STATUS** | | | 0.297 |
| OVERWEIGHT | 1854 (54.50) | 157 (53.40) | |
| OBESE | 1009 (29.70) | 98 (33.30) | |
| **TREATMENT EXPOSURE** | | | <0.001 |
| FULLY EXPOSED | 1075 (45.60) | 51 (24.30) | |
| NON-EXPOSED | 485 (20.60) | 74 (35.20) | |
| PARTIALLY EXPOSED | 798 (33.80) | 85 (40.50) | |

CVE, cardiovascular event; HC, hypercholesterolemia. Data are expressed as the number (%); *In this case, as mean (SD).

CVRFs, treatment exposure was classified as fully exposed in 45.60% of individuals in the group without CVE and in 24.30% of those in the group with CVE.

Results obtained for each of the models, using different groups of variables, are presented below.

### Results obtained for models using cardiovascular risk factors as predictive variables

To facilitate comparison of the predictive capacity of each of the variables tested in XGBoost, RF and NB algorithms, values were normalized on a scale of 0–1 (Fig 2), where 0 and 1 indicate minimum and maximum predictive capacity, respectively.

In all models, the variable that best predicted CVE was age. The next best predictor was physical status in case of the XGBoost and RF models, and HC (followed closely by hypertension) in the case of the NB model.

Table 4 compares the different measures used to evaluate model validity and performance. In terms of F1-score, the best results were obtained for the RF method (0.84). The only parameter for which the XGBoost method outperformed the RF method was specificity (53.00% and 52.05%, respectively). AUC-PR, Log Loss, and F1-Score indicated that the RF method

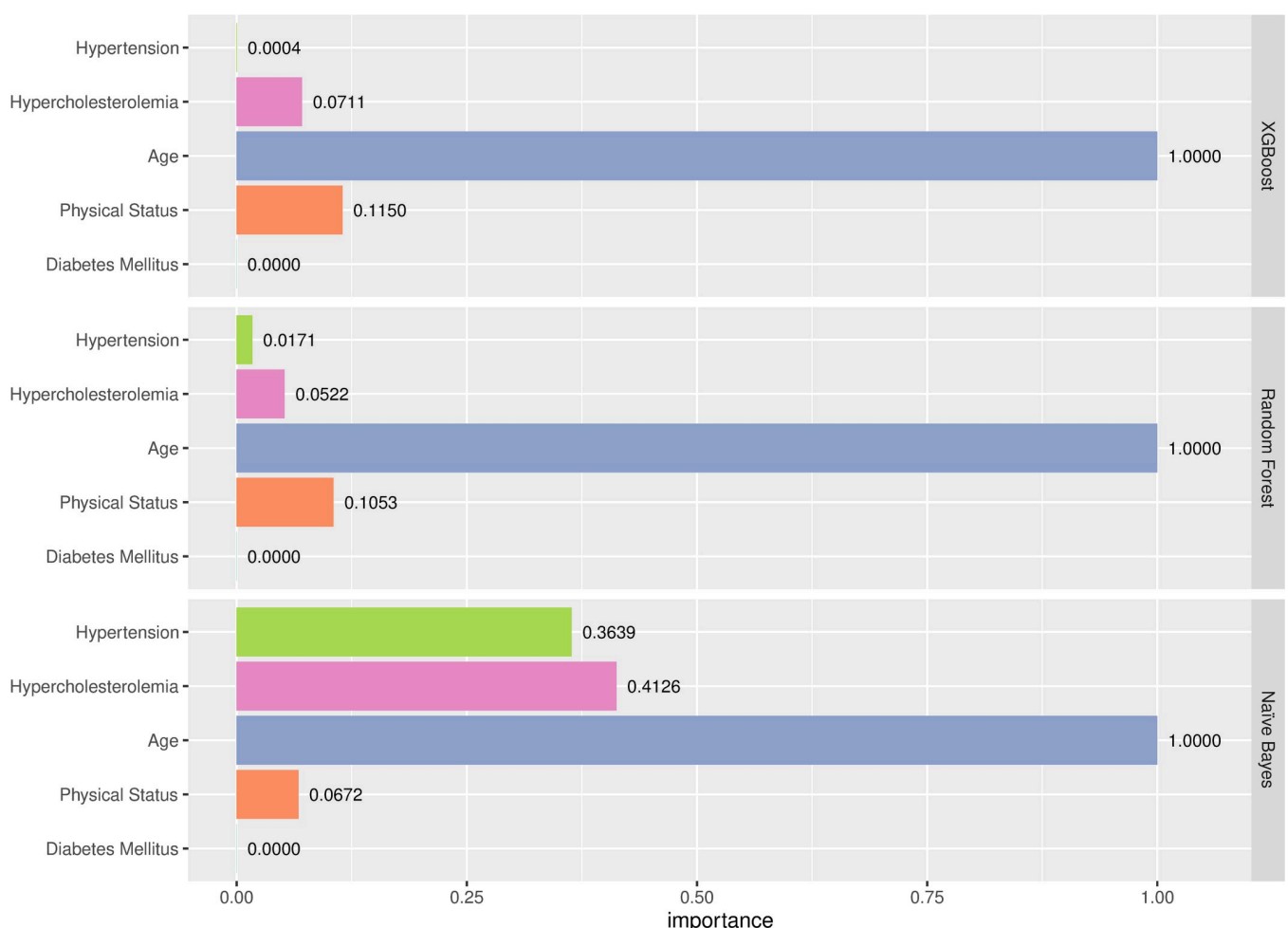

**Fig 2. Predictive capacity of the variables included in the study according to the three algorithms applied: XGBoost, Random Forest and Naïve Bayes.**

**Table 4. Evaluation of model validity and performance using only cardiovascular risk factors as predictive variables.**

| MODEL | ACCURACY (%) | SENSITIVITY (%) | SPECIFICITY (%) | PPV (%) | NPV (%) | AUC-PR | LOG-LOSS | F1-SCORE |
|---|---|---|---|---|---|---|---|---|
| **XGBOOST** | 71.29 | 72.71 | 53.00 | 95.20 | 13.16 | 0.15 | 0.24 | 0.83 |
| **RANDOM FOREST** | 73.96 | 75.66 | 52.05 | 95.29 | 14.30 | 0.17 | 0.24 | 0.84 |
| **NAÏVE BAYES** | 68.29 | 69.96 | 47.00 | 94.42 | 10.88 | 0.11 | 0.26 | 0.80 |

PPV, positive predictable value; NPV, negative predictable value; AUC-PR, Area under the precision-recall curve.

performed best, while the NB method performed the worst (lowest AUC-PR and F1-Score and highest Log Loss).

## Results obtained for models using cardiovascular risk factors and treatment exposure as predictive variables

When treatment exposure was also included as a predictive variable (Fig 3), age remained the variable that best predicted in both the XGBoost and RF models, followed by treatment

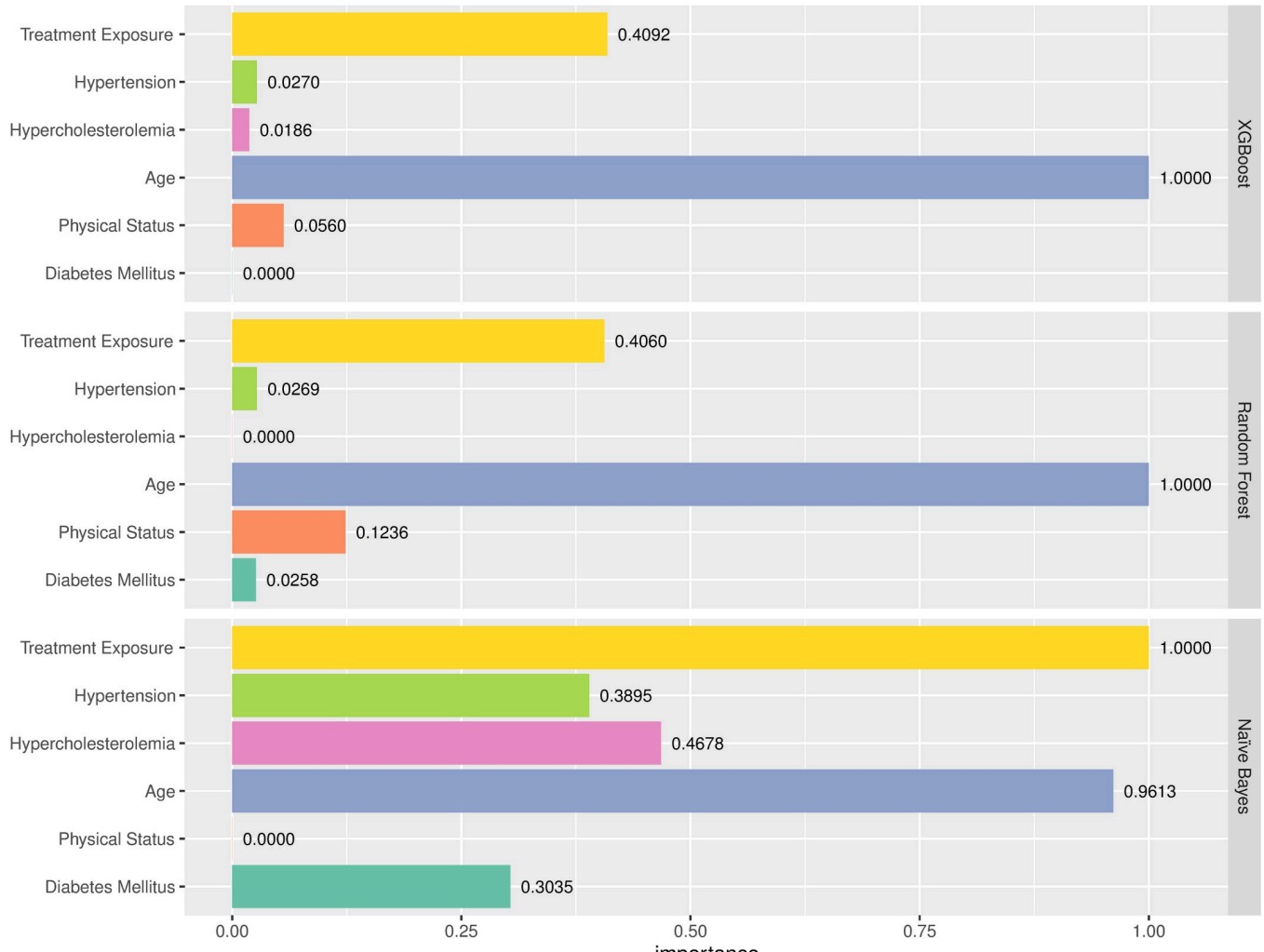

**Fig 3. Predictive capacity of the variables included in the study in each of the three algorithms applied: XGBoost, Random Forest, and Naïve Bayes.**

**Table 5. Evaluation of model validity and performance using both cardiovascular risk factors and treatment exposure as predictive variables.**

| MODEL | ACCURACY (%) | SENSITIVITY (%) | SPECIFICITY (%) | PPV (%) | NPV (%) | AUC-PR | LOG-LOSS | F1-SCORE |
|---|---|---|---|---|---|---|---|---|
| XGBOOST | 72.39 | 73.17 | 63.36 | 95.85 | 16.94 | 0.24 | 0.25 | 0.83 |
| RANDOM FOREST | 73.35 | 73.68 | 69.52 | 96.55 | 18.57 | 0.28 | 0.24 | 0.84 |
| NAÏVE BAYES | 61.78 | 61.88 | 60.62 | 94.79 | 12.07 | 0.13 | 0.28 | 0.75 |

PPV, positive predictable value; NPV, negative predictable value; AUC-PR, Area under the precision-recall curve.

exposure. In both these models all other variables had very little influence, although physical status had a higher predictive capacity in the RF versus the XGBoost model. In the NB model treatment exposure was the variable with the highest predictive capacity, followed closely by age.

Table 5 compares the different parameters used to evaluate model validity and performance. The RF model showed the highest scores for accuracy, sensitivity, specificity, PPV, and NPV, while the NB model showed the lowest scores for these parameters. Similar results were shown by the tests calculated to evaluate the effectiveness of the models: the best results were for the RF algorithm and the worst for the NB, being the XGBoost scores similar to the RF ones.

## Comparison of the models

The RF model performed best, regardless of the groups of variables included. Fig 4 shows the PR curve for this method when considering both CVRFs and treatment exposure as predictive variables. The model performed best when both groups of variables were included.

Log Loss and F1 score were very similar regardless of the variables included (0.24 and 0.84 for CVRFs and CVRFs + treatment exposure, respectively). Parameters used to evaluate model validity (accuracy, sensitivity, and PPV) were very similar in both models (about 73%, 74%, and 96%). However, specificity and NPV were considerably higher when treatment exposure was included as a predictive variable (69.52% and 18.57%, respectively, versus 52.05% and 14.30%, respectively, when treatment exposure was excluded).

Finally, age and treatment exposure had the highest predictive capacity. Age was the variable that best predicted CVE in all models, except in the NB model when both CVRFs and treatment exposure were included as predictive variables: in that scenario, treatment exposure was the most important predictor of CVE, followed closely by age. When treatment exposure was excluded from the RF and XGBoost models, age was the variable with the greatest predictive capacity. While the predictive capacity of age and treatment exposure were 1 and 0.40, respectively, in both the RF and XGBoost models, the corresponding values in the NB model were much closer to one another (0.96 and 1, respectively). In the NB model, after age, the variables HC and hypertension had greater predictive capacity than physical status and DM.

Evaluation of model performance showed that the RF model performed best, regardless of the variables included, followed by XGBoost. Both models performed better when CVRFs and treatment exposure were included as predictive variables, compared with CVRFs alone (Fig 4).

## Discussion

In the present study we compared the CVE prediction performance of three machine learning algorithms, in each case using two distinct combinations of predictive variables: (i) CVRFs (age, physical status, HC, hypertension and DM); and (ii) CVRFs plus treatment exposure.

Age was the variable with the greatest capacity to predict CVE in all models, except for the NB model when CVRFs + treatment exposure were included as predictive variables, in which

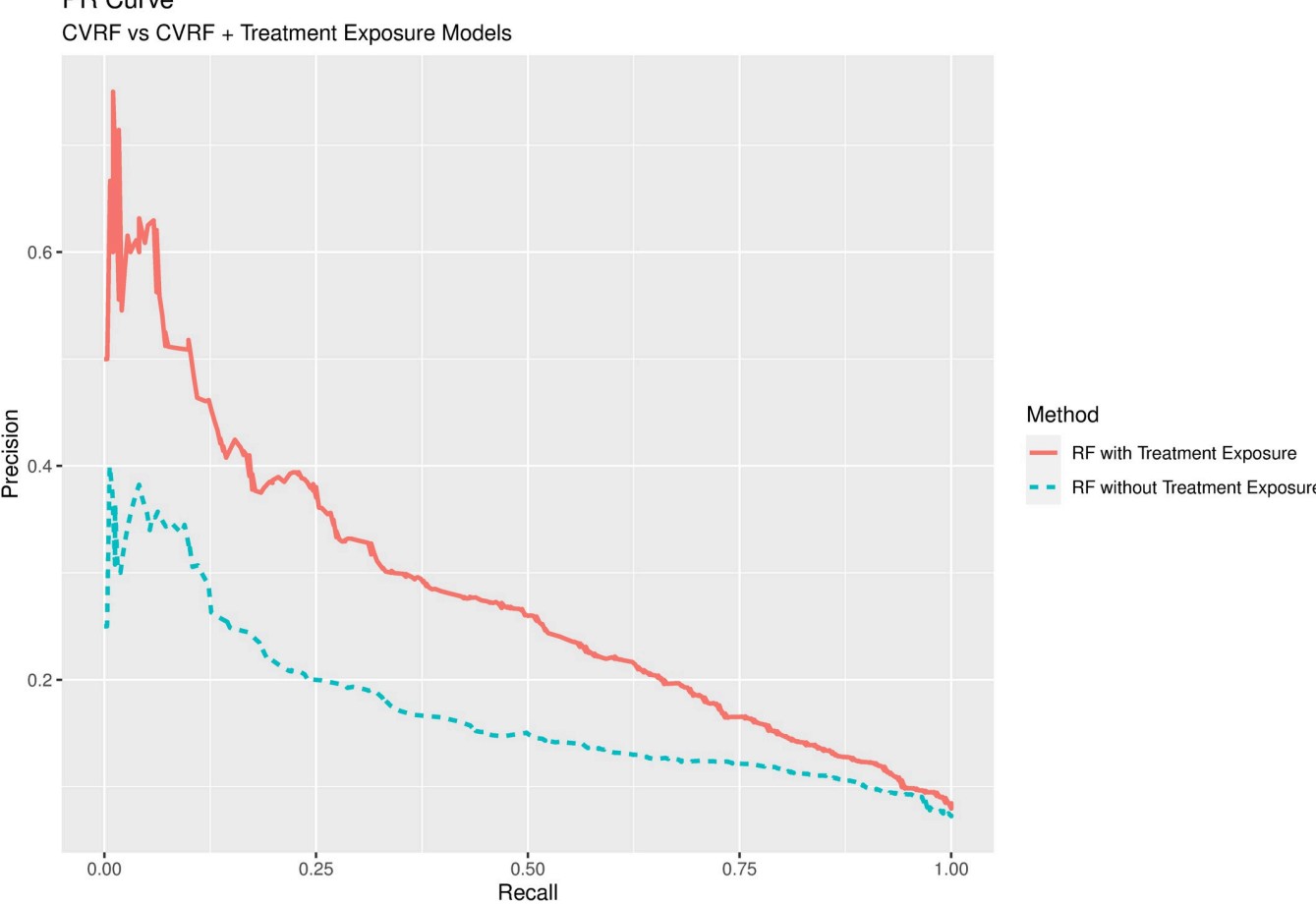

**Fig 4. PR curves obtained for RF model, considering cardiovascular risk factors alone or cardiovascular risk factors and treatment exposure as predictive variables.**

case treatment exposure showed the greatest predictive capacity, followed closely by age. When only CVRFs were included, age showed the greatest predictive capacity, while that of other CVRFs was markedly lower, and varied depending on the model used. When treatment exposure was added as an additional predictive variable, its predictive capacity was closer to that of age than any of the other CVRFs. These findings indicate that treatment exposure plays an important role in determining the incidence of CVE and should therefore be taken into account when managing cardiovascular risk.

As said before, age was the variable that best predicted CVE occurrence, while the predictive capacity of other CVRFs differed across models. The relationship between CVD and age, HC, hypertension, DM, and physical status is well documented [26–29]. Studies that included age as a predictive variable have unanimously shown that this parameter has the greatest predictive power, suggesting that age is a key CVRF [3, 4, 12, 26]. Furthermore, the combination of two or more CVRFs increases the risk of mortality [29], although the influence of individual CVRFs varies between studies [17–19]. In their study of the prevalence of CVRFs in individuals who experienced a CVE, Yandrapalli et al. found that HC was the most prevalent, followed closely by hypertension, smoking, and finally DM [27]. Another study [29] found that hypertension was the CVRF most closely associated with all-cause mortality, followed by DM, HC, and overweight. Huang et al. [28] concluded that the most important CVRFs were obesity and

smoking in rural and urban areas, respectively, followed by dyslipidemia. Thus, apart from age, it remains unclear which other CVRF is the greatest determinant of the incidence of CVE.

Proper pharmacological control of modifiable CVRFs is essential to reduce the risk of a CVE. Clinical guidelines propose the objectives to control CVRFs in order to decrease this risk [30, 31]. Previous research [32–34] has shown that adherence to these treatments is suboptimal, and the methods most commonly used to determine the risk of CVE do not include treatment adherence as a predictive variable. There is also evidence [35] that a considerable number of CVEs are due to poor adherence to cardiovascular preventive treatments. Therefore, measuring adherence could maximize the potential of effective cardiac therapies in clinical settings. Exposure to drugs for control of CVRFs could be an important factor to consider when determining the risk of experiencing a CVE, as treatment exposure varies considerably between individuals and its plays a key role in risk management. This view is borne out in our study, in which model performance improved considerably when treatment exposure (determined based on the adherence to each individual treatment) was included as a predictive variable.

Evidence indicates that population-level screening for CVD risk and CVRFs, including screenings performed in a work setting, is not effective in decreasing CVD morbidity and mortality [36]. However, primary-care-based interventions targeting individuals at high risk, based on their age or risk factors, seem to be effective [36]. The models presented in the present study could be applied in clinical practice to assess the individual risk of CVE based on patient characteristics and treatment adherence, and could therefore fulfil this screening role. Furthermore, they can help orient the intervention and identify the most appropriate measures to take (e.g. behavioral change versus reinforcement of adherence).

Over the years, much effort has been invested in calculating CVE risk, using a variety of methods, many of which have limitations that can be overcome with machine learning technics. These techniques offer a variety of approaches to process huge amounts of data to predict the incidence of CVE, thus allowing researchers and clinicians to select the algorithm that better suits their data or objectives.

Previous studies [5, 37] comparing different algorithms found that the RF model provided the most accurate results, in line with our findings. A previous comparison of XGBoost and NB [12] found that XGBoost performed better, as also observed in the present study. Finally, when comparing performance of RF and XGBoost algorithms with SCORE and Framingham risk score, the first ones had better results [5].

This study has some limitations, data were highly imbalanced and some methods were applied to deal with that. Because of that, AUC-PR values were low. In addition, the NPV value was low for all models, this could mean that a certain factor has been omitted from the models. Furthermore, some of the CVEs included in the study (e.g. arrhythmias) may be unrelated to the CVRFs considered. Nonetheless, our study revealed good performance for all the models, in particular RF and XGBoost when treatment exposure was included as a predictive variable. Other limitations, related to the kind of data available, include the absence of smoking data for the entire study period, since smoking is one of the most important modifiable risk factors for CVE, as well as the absence of women in the cohort, as sex also influences CVE risk.

This study has several strengths. First is the use of three different machine learning techniques, which integrate all available data and offer several advantages over earlier algorithms, as explained above, and compare the results between them to determine which is more accurate. In the context of machine learning studies, to our knowledge this is the first study to consider these specific groups of variables in the prediction of CVD. Another key strength is the inclusion of adherence to different treatments integrated into a single variable: this is also the first study to use this approach to predict CVE, and we consider this approach essential to

determine the true risk of experiencing a CVE, given the influence of adherence on therapeutic outcome. Finally, our analysis considered multiple algorithms and different combinations of predictive variables, allowing us to identify the model that performed best in this particular study population and to evaluate the influence of different variables on CVE occurrence.

Further studies that consider additional CVRFs, including smoking habits, and to include more heterogeneous populations to better reflect the situation in terms of presence of CVRF and CV pathology will be needed to evaluate the contribution of CVRF to CVE risk. Moreover, it will be essential to include women in future studies given the differences in the incidence and relevance of CVRFs between sexes.

## Conclusions

We found that the age was the most influential variable to predict the occurrence of a CVE followed by the treatment exposure. The rest of variables considered changed its importance depending on the algorithm and the model implemented. The use of machine learning techniques can be of great help to assess the risk of suffering a CVE including a huge amount of data and can be applied for personalized medicine to prevent CVE. The usefulness of machine learning techniques has been proven and the algorithm that better results gave in our case was RF, that improved its results adding the treatment exposure as variable. This study brings to light the importance of considering treatment exposure, estimated based on the adherence to therapy, when trying to assess the risk of suffering from a CVE.

## Author Contributions

**Conceptualization:** Sara Castel-Feced, Lina Maldonado, María José Rabanaque-Hernández.

**Data curation:** Sara Castel-Feced, Sara Malo, Isabel Aguilar-Palacio, Cristina Feja-Solana, José Antonio Casasnovas.

**Formal analysis:** Sara Castel-Feced, Lina Maldonado, María José Rabanaque-Hernández.

**Funding acquisition:** Sara Malo, Isabel Aguilar-Palacio.

**Methodology:** Sara Castel-Feced, Lina Maldonado, María José Rabanaque-Hernández.

**Supervision:** María José Rabanaque-Hernández.

**Writing – original draft:** Sara Castel-Feced, María José Rabanaque-Hernández.

**Writing – review & editing:** Sara Castel-Feced, Sara Malo, Isabel Aguilar-Palacio, Cristina Feja-Solana, José Antonio Casasnovas, Lina Maldonado, María José Rabanaque-Hernández.

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
