## [Decision Letter · Decision Letter 0]

7 Aug 2023

PONE-D-23-06851Influence of cardiovascular risk factors and treatment exposure on cardiovascular event incidence: assessment using machine learning algorithmsPLOS ONE

Dear Dr. Castel Feced,

Thank you for submitting your manuscript to PLOS ONE. After careful consideration, we feel that it has merit but does not fully meet PLOS ONE’s publication criteria as it currently stands. Therefore, we invite you to submit a revised version of the manuscript that addresses the points raised during the review process. As you work on the revisions, please keep in mind the importance of responding to Reviewer 1's comments in detail, and consider how implementing their suggestions can elevate your work.

We look forward to receiving your revised manuscript.

Kind regards,

Chi-Shin Wu

Academic Editor

PLOS ONE

Journal Requirements:

Reviewers' comments:

Reviewer's Responses to Questions

**Comments to the Author**

1. Is the manuscript technically sound, and do the data support the conclusions?

Reviewer #1: No

Reviewer #2: Yes

2. Has the statistical analysis been performed appropriately and rigorously? 

Reviewer #1: No

Reviewer #2: Yes

3. Have the authors made all data underlying the findings in their manuscript fully available?

Reviewer #1: Yes

Reviewer #2: Yes

4. Is the manuscript presented in an intelligible fashion and written in standard English?

Reviewer #1: Yes

Reviewer #2: Yes

5. Review Comments to the Author

Reviewer #1: Thank you for the opportunity to read this fascinating article. Nonetheless, I would like to make a few suggestions.

The introduction is extremely lengthy. Please trim it down, particularly lines 71 to 81. The authors could instead provide a reference to these well-known algorithms and, more importantly, examples of their use in CVD risk prediction. Justify the algorithm selections for machine learning

Line 64 – 66 As LIME or SHAP analysis exists to overcome the black box nature of the machine learning algorithm, it appears that the authors' claims are irrelevant. Consider including either method in this study.

SMOTE is one example of balancing the data up and down to achieve balance should consider using it.

Please refer, between lines 82 and 83, to papers that employ the Machine Learning algorithm for CVD risk estimation instead of other diseases.

Please rewrite the purpose of this study and specify which algorithm is being employed. Please also consider including a comparison with the conventional FRS score. Please specify the limitation of the FRS. Has it been validated in your population? If so, what are the outcomes?

Methods

Provide a flowchart and describe the total number of participants from various data sources. Include the percentage of missing data for each variable utilised in this study.

Since data is randomly divided, are there duplicate records for the same patients?

Line 198 – 206 The authors' claims that the data is separated into training and testing data and that the entire data set is also used for testing are unclear. If the number of datasets is low, it is recommended to use K fold. Moreover, what measures are taken to address data imbalances? Is it rational to change the threshold value to 0.1? What scientific evidence supports this claim? Provide references for justifications of any steps.

The data normalisation at lines 259m to 261 should be placed in the methods section under data pre processing. I recommend that the authors rewrite the methods section and include a flowchart to facilitate comprehension.

AUC, not Accuracy, should be the primary performance metric for evaluating medical dataset models on lines 267 to 274. Clarify the AUC PR and F1-score. Please report the AUC value for tables 3 and 4, as it should be used to compare models. Include a comparison to the FRS score as well. Compare with other ML papers on CVD risk for the purpose of benchmarking.

I recommend that the authors implement variable importance, which can be generated using RF and XGB, in order to determine which features are of greater importance. Compare the results of the ML model to those of statistical analysis.

Reviewer #2: In the present work the authors have used several machine learning methods for the prediction of cardiovascular event incidence using two sets of features: cardiovascular risk factors and treatment exposure. They developed separate models and also combined models using these features and reported better prediction accuracies using combined feature set. I have few comments which are given beklow

Comments-

1. The authors should mention how the predictive capacity for the features are being deduced?

2. There are no p-values for the overweight and obese features.

6. PLOS authors have the option to publish the peer review history of their article (what does this mean?). If published, this will include your full peer review and any attached files.

Reviewer #1: **Yes: **Sorayya Malek

Reviewer #2: No

---

## [Author Response · Author response to Decision Letter 0]

20 Sep 2023

Reviewer #1: Thank you for the opportunity to read this fascinating article. Nonetheless, I would like to make a few suggestions.

We want to thank the reviewer her suggestions, which have been useful to improve the quality and clarity of the manuscript. We proceed to answer, point by point, his/her comments:

1. The introduction is extremely lengthy. Please trim it down, particularly lines 71 to 81. The authors could instead provide a reference to these well-known algorithms and, more importantly, examples of their use in CVD risk prediction. 

Thank you for your comment. As suggested by the reviewer, we have reduced the introduction, and the paragraph suggested now is as follows (lines 73-76):

“There are different kinds of supervised machine learning. When the variable to be predicted is categorical (e.g. a cardiovascular event), classification machine learning techniques [3,4] such as Naïve Bayes (NB) algorithms and ensemble methods are used [12,14,15]. Ensemble methods include bagging and boosting methods such are Random Forest (RF), XG Boost [12,14,15]. “

2. Justify the algorithm selections for machine learning.

To justify the algorithms selected in the present study, we have added some references in which these algorithms are applied to predict cardiovascular disease (lines 73-76):

“When the variable to be predicted is categorical (e.g. a cardiovascular event), classification machine learning techniques [3,4] such as Naïve Bayes (NB) algorithms and ensemble methods are used [12,14,15]. Ensemble methods include bagging and boosting methods such are Random Forest (RF), XG Boost[12,14,15].”

References:

 3. Alaa AM, Bolton T, Angelantonio E di, Rudd JHF, van der Schaar M. Cardiovascular disease risk prediction using automated machine learning: A prospective study of 423,604 UK Biobank participants. PLoS One [Internet]. 2019 May 1 [cited 2021 May 14];14(5). Available from: https://pubmed.ncbi.nlm.nih.gov/31091238/

 4. Ambale-Venkatesh B, Yang X, Wu CO, Liu K, Gregory Hundley W, McClelland R, et al. Cardiovascular Event Prediction by Machine Learning: The Multi-Ethnic Study of Atherosclerosis. Circ Res. 2017 Oct 13;121(9):1092–101. 

 12. Wang K, Tian J, Zheng C, Yang H, Ren J, Liu Y, et al. Interpretable prediction of 3-year all-cause mortality in patients with heart failure caused by coronary heart disease based on machine learning and SHAP. Comput Biol Med. 2021 Oct 1;137:104813. 

 14. Tsarapatsani K, Sakellarios AI, Pezoulas VC, Tsakanikas VD, Kleber ME, Marz W, et al. Machine Learning Models for Cardiovascular Disease Events Prediction. Annu Int Conf IEEE Eng Med Biol Soc [Internet]. 2022 [cited 2023 Sep 13];2022:1066–9. Available from: https://pubmed.ncbi.nlm.nih.gov/36085658/

 15. Garavand A, Salehnasab C, Behmanesh A, Aslani N, Zadeh AH, Ghaderzadeh M. Efficient Model for Coronary Artery Disease Diagnosis: A Comparative Study of Several Machine Learning Algorithms. J Healthc Eng. 2022. 

3. Line 64 – 66 As LIME or SHAP analysis exists to overcome the black box nature of the machine learning algorithm, it appears that the authors' claims are irrelevant. Consider including either method in this study.

Thank you for your comment. We analyse the importance of features applying caret package in R (Kuhn M, Building Predictive Models in R Using the caret Package). To clarify it we added some lines in the introduction and in the methods as follows:

Introduction (lines 67-68):

“This problem has been overcome by the development of different methods, implemented in different R libraries, that try to set the importance of each feature in the prediction [10–12].

Methodology (lines 231-243):

“For RF models, the method applied to compute the contribution of each variable consisted of recoding for each tree the prediction accuracy on the out-of-bag portion. Then each predictor variable was permuted and the same was done. Finally, for all trees, the difference between both accuracies was averaged and normalized by the standard score [10,11]. 

For XG Boost models, the reduction in the loss function attributable to each variable in sum of squared error in predicting the gradient on each iteration was calculated. Finally, the improvement score for each predictor was averaged across all the trees in the ensemble [11,25].

Finally, for models developed applying NB a ROC curve analysis was conducted on each predictor. Different cutoffs were applied to the predictor data to predict the class. Then, sensitivity and specificity were computed for each cutoff and the area under ROC curve was calculated using the trapezoidal rule. This area was used as the measure of variable importance[11].”

References:

 10. Greenwell BM, Boehmke BC. Variable Importance Plots-An Introduction to the vip Package. 

 11. Kuhn M. Building Predictive Models in R Using the caret Package. J Stat Softw [Internet]. 2008;28(5). Available from: http://www.jstatsoft.org/

 12. Wang K, Tian J, Zheng C, Yang H, Ren J, Liu Y, et al. Interpretable prediction of 3-year all-cause mortality in patients with heart failure caused by coronary heart disease based on machine learning and SHAP. Comput Biol Med. 2021 Oct 1;137:104813.

 25. Jiaming Yuan. Package ‘xgboost’. 2023 [cited 2023 Sep 14]; Available from: https://github.com/dmlc/xgboost/issues

4. SMOTE is one example of balancing the data up and down to achieve balance should consider using it.

Thank you for the comment. According to the literature (Haibo He et al., Imbalanced Learning: Foundations, Algorithms, and Applications; Maloof MA, Learning When Data Sets are Imbalanced and When Costs are Unequal and Unknown; Collell G et al, Reviving Threshold-Moving: a Simple Plug-in Bagging Ensemble for Binary and Multiclass Imbalanced Data), there are different methods to deal with imbalanced data, in this study it was applied a threshold-moving and the parameter chosen to select the best threshold was the F1 score ( Miao J et al., Precision–recall curve (PRC) classification trees). Further explanations and bibliography will be given in comment number 11. 

5. Please refer, between lines 82 and 83, to papers that employ the Machine Learning algorithm for CVD risk estimation instead of other diseases.

As suggested, we added some examples of manuscripts that use machine learning techniques to predict cardiovascular disease (line 77-78):

“Machine learning techniques represent a promising approach for CVD risk prediction given their demonstrated potential [3,4,12,14–16] and the advantages they offer over existing scoring systems.”

References:

 3. Alaa AM, Bolton T, Angelantonio E Di, Rudd JHF, van der Schaar M. Cardiovascular disease risk prediction using automated machine learning: A prospective study of 423,604 UK Biobank participants. PLoS One [Internet]. 2019 May 1;14(5). Available from: https://pubmed.ncbi.nlm.nih.gov/31091238/

 4. Ambale-Venkatesh B, Yang X, Wu CO, Liu K, Gregory Hundley W, McClelland R, et al. Cardiovascular Event Prediction by Machine Learning: The Multi-Ethnic Study of Atherosclerosis. Circ Res. 2017 Oct 13;121(9):1092–101.

 12. Wang K, Tian J, Zheng C, Yang H, Ren J, Liu Y, et al. Interpretable prediction of 3-year all-cause mortality in patients with heart failure caused by coronary heart disease based on machine learning and SHAP. Comput Biol Med. 2021 Oct 1;137:104813. 

 14. Tsarapatsani K, Sakellarios AI, Pezoulas VC, Tsakanikas VD, Kleber ME, Marz W, et al. Machine Learning Models for Cardiovascular Disease Events Prediction. Annu Int Conf IEEE Eng Med Biol Soc [Internet]. 2022 [cited 2023 Sep 13];2022:1066–9. Available from: https://pubmed.ncbi.nlm.nih.gov/36085658/

 15. Garavand A, Salehnasab C, Behmanesh A, Aslani N, Zadeh AH, Ghaderzadeh M. Efficient Model for Coronary Artery Disease Diagnosis: A Comparative Study of Several Machine Learning Algorithms. J Healthc Eng. 2022;2022. 

 16. Aziz F, Malek S, Ibrahim KS, Shariff RER, Wan Ahmad WA, Ali RM, et al. Short- and long-term mortality prediction after an acute ST-elevation myocardial infarction (STEMI) in Asians: A machine learning approach. PLoS One [Internet]. 2021 Aug 1 [cited 2023 Sep 14];16(8):e0254894. Available from: https://journals.plos.org/plosone/article?id=10.1371/journal.pone.0254894

6. Please rewrite the purpose of this study and specify which algorithm is being employed.

As suggested by the reviewer, we added the algorithms in the objectives (lines 78-81):

“In the present study, we analyzed the ability of three machine learning algorithms, NB, RF and XG Boost, to predict the appearance of cardiovascular events (CVE) and analysed the influence of different CVRF-related variables on CVE prediction.”

7. Please also consider including a comparison with the conventional FRS score. Please specify the limitation of the FRS. Has it been validated in your population? If so, what are the outcomes?

FRS was calibrated for a Spanish population (Marrugat J et al, Estimación del riesgo coronario en España mediante la ecuación de Framingham calibrada) and it has been validated in other Spanish cohorts showing different results (Ruiz-Villaverde G, et al.,Agreement between Framingham-DORICA and SCORE scales in estimation of cardiovascular risk in the patients suffering from metabolic syndrome in Granada; Amor AJ et al., Prediction of Cardiovascular Disease by the Framingham-REGICOR Equation in the High-Risk PREDIMED Cohort: Impact of the Mediterranean Diet Across Different Risk Strata; Buitrago F, et al. Original and REGICOR Framingham functions in a nondiabetic population of a Spanish health care center: a validation study). Nevertheless, one of the objectives of the present study was to analise the role of adherence to treatment in the prediction of CVE, and FRS score does not consider this information. Furthermore, some of the variables included in the FRS were not available during the years that the study was conducted as it is explained in the limitations as follows (line 394-397):

“Other limitations, related to the kind of data available, include the absence of smoking data for the entire study period, since smoking is one of the most important modifiable risk factors for CVE.”

Finally, we included a sentence about the comparison of machine learning techniques and FRS in the Discussion (lines 385-386) as follows:

“Finally, when comparing performance of RF and XGBoost algorithms with SCORE and Framingham risk score, the first ones had better results [5]. “

Reference:

 5. Jamthikar AD, Gupta D, Mantella LE, Saba L, Laird JR, Johri AM, et al. Multiclass machine learning vs. conventional calculators for stroke/CVD risk assessment using carotid plaque predictors with coronary angiography scores as gold standard: a 500 participants study. Int J Cardiovasc Imaging. 2021 Apr 1;37(4):1171–87. doi:10.1007/S10554-020-02099-7/TABLES/5

Methods

8. Provide a flowchart and describe the total number of participants from various data sources. Include the percentage of missing data for each variable utilised in this study.

Thank you for your comment. To clarify the selection of participants and the model development procedure we introduced a flowchart in the methods section. To explain how variables were extracted, its origin and the number of missing data, we did a table and we introduced both of them in the methods section as follows: 

Line 96: “The selection of patients is explained in Figure 1.

Fig 1: Flowchart depicting the study population and model development." (Figure available in the response to reviewers word file)

Line 115: 

The source of the variables and number of missing data is summarised in Table 1.

Table 1: Variables sources and missing data. (Table available in the manuscript file and in the response to reviewers word file)

9. Since data is randomly divided, are there duplicate records for the same patients?

Thank you for your comment. It was applied a random sampling without replacement, so there were not duplicate records for any patient.

10. Line 198 – 206 The authors' claims that the data is separated into training and testing data and that the entire data set is also used for testing are unclear. If the number of datasets is low, it is recommended to use K fold. 

As it is explained in lines 202-205, 5 fold and 10 fold stratification was applied to train models and to adjust the parameters in the different models. To testing the models all the sample was considered.

“To fit the hyperparameters and avoid over-fitting, the prediction accuracy of all models was tested using 5 and 10 fold stratified cross validation (cv) to estimate F1-score. Results were similar when applying 5 and 10 fold cv, so results shown in the article corresponds to the 5-fold cv.”

11. Moreover, what measures are taken to address data imbalances? Is it rational to change the threshold value to 0.1? What scientific evidence supports this claim? Provide references for justifications of any steps.

Thank you for your comment. To address data imbalanced we have applied a threshold-moving attending to the F1-measure. After analyzing different thresholds to evaluate which maximizes F1-measure, it pointed to 0.1 as the best threshold in all the models computed. We added a section in the methodology to explain how the models assessment was done and added some references (lines 219-226):

“Machine learning models assessment

Because of the highly imbalanced data that we had, to validate models threshold were moved and selected based on max of f1 score in P-R curves, being 0.1 in all models. To measure the validity of the models, the following measures were taken into account: accuracy, sensitivity, specificity, positive predictive value (PPV) and negative predictive value (NPV). Next, three different tests were applied to evaluate the performance of the model: area under the precision-recall curve (AUC-PR), Log Loss, and F1 score. These scores were selected because different studies recommend them for imbalanced data [22–24].”

References: 

 22. Gaudreault JG, Branco P, Gama J. An Analysis of Performance Metrics for Imbalanced Classification. Lecture Notes in Computer Science (including subseries Lecture Notes in Artificial Intelligence and Lecture Notes in Bioinformatics) [Internet]. 2021 [cited 2023 Sep 15];12986 LNAI:67–77. Available from: https://link.springer.com/chapter/10.1007/978-3-030-88942-5_6

 23. Miao J, Zhu W. Precision–recall curve (PRC) classification trees. Evol Intell [Internet]. 2022 Sep 1 [cited 2023 Sep 15];15(3):1545–69. Available from: https://link.springer.com/article/10.1007/s12065-021-00565-2

 24. Bekkar M, Djemaa HK, Alitouche TA. Evaluation Measures for Models Assessment over Imbalanced Data Sets. Journal of Information Engineering and Applications [Internet]. 2013;3(10):27–38. Available from: http://www.iiste.org/Journals/index.php/JIEA/article/view/7633

12. The data normalisation at lines 259m to 261 should be placed in the methods section under data pre processing. I recommend that the authors rewrite the methods section and include a flowchart to facilitate comprehension.

The normalization in these lines refers to the features importance scores. The method to compute scores for features importance was different for each algorithm and, because of that, the range of the scores differ among them. To facilitate the comparability of the features importance, we normalize these scores between 0 and 1 as in other studies. We have clarified it in lines 228-231:

“After achieving valid and accurate models, the contribution of each variable to the prediction was extracted using caret R package. Methods applied for each algorithm are different and they give scores in different ranges so, to facilitate comparability, scores obtained for each method were normalized to a scale of 0–1.”

As suggested by the reviewer, we rewrite some parts of the methodology, specially the statistical analysis part and we divided it in 3 different sections. 

• Machine learning models development (line 196)

• Machine learning models assessment (line 219)

• Variables importance (line 227)

Furthermore, in the flowchart, Figure 1 (shown in response to the reviewers word file), some specifications of the model development

process were depicted.

Line 197:

“Supervised machine learning algorithms were used to determine the utility of different variables to predict CVE. The process is depicted in Figure 1.”

13. AUC, not Accuracy, should be the primary performance metric for evaluating medical dataset models on lines 267 to 274. Clarify the AUC PR and F1-score. Please report the AUC value for tables 3 and 4, as it should be used to compare models. Include a comparison to the FRS score as well. Compare with other ML papers on CVD risk for the purpose of benchmarking.

As suggested by the reviewer and following different recommendations (Gaudreault JG, et al., An Analysis of Performance Metrics for Imbalanced Classification; Miao J et al., Precision–recall curve (PRC) classification trees) for evaluation measures in studies with imbalanced data, we changed these lines (now lines 282-287) as follows:

“Table 4 compares the different measures used to evaluate model validity and performance. In terms of F1-score, the best results were obtained for the RF method (0.84). The only parameter for which the XGBoost method outperformed the RF method was specificity (53.00% and 52.05%, respectively). AUC-PR, Log Loss, and F1-Score indicated that the RF method performed best, while the NB method performed the worst (lowest AUC-PR and F1-Score and highest Log Loss).”

To clarify the AUC PR and F1-score we added the following sentence and add some references in lines 225-226: 

“These scores were selected because different studies recommend them for imbalanced data [22–25].”

References: 

 22. Miao J, Zhu W. Precision–recall curve (PRC) classification trees. Evol Intell [Internet]. 2022 Sep 1 [cited 2023 Sep 15];15(3):1545–69. Available from: https://link.springer.com/article/10.1007/s12065-021-00565-2

 24. Bekkar M, Djemaa HK, Alitouche TA. Evaluation Measures for Models Assessment over Imbalanced Data Sets. Journal of Information Engineering and Applications [Internet]. 2013;3(10):27–38. Available from: http://www.iiste.org/Journals/index.php/JIEA/article/view/7633

 25. Jiaming Yuan. Package ‘xgboost’. 2023 [cited 2023 Sep 14]; Available from: https://github.com/dmlc/xgboost/issues

Area under the curve for PR is shown in tables 3 and 4 (now tables 4 and 5) in 7th column (Tables are in lines 288 and 306 in the manuscript and in the response to reviewers word file)

14. I recommend that the authors implement variable importance, which can be generated using RF and XGB, in order to determine which features are of greater importance. Compare the results of the ML model to those of statistical analysis.

Thank you for your comment, as explained in previous commentaries, the feature analysis was conducted using caret package. To clarify it we have added one section in the methodology about variable importance and we expand the explanation of how it was done as follows (lines 227-243)

“After achieving valid and accurate models, the contribution of each variable to the prediction was extracted using caret R package. Methods applied for each algorithm are different and they give scores in different ranges so, to facilitate comparability, scores obtained for each method were normalized to a scale of 0–1. For RF models, the method applied to compute the contribution of each variable consisted of recoding for each tree the prediction accuracy on the out-of-bag portion. Then each predictor variable was permuted and the same was done. Finally, for all trees, the difference between both accuracies was averaged and normalized by the standard score [10,11]. 

For XG Boost models, the reduction in the loss function attributable to each variable in sum of squared error in predicting the gradient on each iteration was calculated. Finally, the improvement score for each predictor was averaged across all the trees in the ensemble [11,25].

Finally, for models developed applying NB a ROC curve analysis was conducted on each predictor. Different cutoffs were applied to the predictor data to predict the class. Then, sensitivity and specificity were computed for each cutoff and the area under ROC curve was calculated using the trapezoidal rule. This area was used as the measure of variable importance[11].”

References:

 10. Greenwell BM, Boehmke BC. Variable Importance Plots-An Introduction to the vip Package. 

 11. Kuhn M. Building Predictive Models in R Using the caret Package. J Stat Softw [Internet]. 2008;28(5). Available from: http://www.jstatsoft.org/

 12. Wang K, Tian J, Zheng C, Yang H, Ren J, Liu Y, et al. Interpretable prediction of 3-year all-cause mortality in patients with heart failure caused by coronary heart disease based on machine learning and SHAP. Comput Biol Med. 2021 Oct 1;137:104813.

 25. Jiaming Yuan. Package ‘xgboost’. 2023 [cited 2023 Sep 14]; Available from: https://github.com/dmlc/xgboost/issues

 

Reviewer #2: In the present work the authors have used several machine learning methods for the prediction of cardiovascular event incidence using two sets of features: cardiovascular risk factors and treatment exposure. They developed separate models and also combined models using these features and reported better prediction accuracies using combined feature set. I have few comments which are given beklow

Comments-

1. The authors should mention how the predictive capacity for the features are being deduced?

Thank you for your comment. We analyse the importance of features applying caret package in R. To clarify it we added some lines in the methods as follows:

Methodology (lines 231-243):

“For RF models, the method applied to compute the contribution of each variable consisted of recoding for each tree the prediction accuracy on the out-of-bag portion. Then each predictor variable was permuted and the same was done. Finally, for all trees, the difference between both accuracies was averaged and normalized by the standard score [10,11]. 

For XG Boost models, the reduction in the loss function attributable to each variable in sum of squared error in predicting the gradient on each iteration was calculated. Finally, the improvement score for each predictor was averaged across all the trees in the ensemble [11,25].

Finally, for models developed applying NB a ROC curve analysis was conducted on each predictor. Different cutoffs were applied to the predictor data to predict the class. Then, sensitivity and specificity were computed for each cutoff and the area under ROC curve was calculated using the trapezoidal rule. This area was used as the measure of variable importance[11].”

References:

 10. Greenwell BM, Boehmke BC. Variable Importance Plots-An Introduction to the vip Package. 

 11. Kuhn M. Building Predictive Models in R Using the caret Package. J Stat Softw [Internet]. 2008;28(5). Available from: http://www.jstatsoft.org/

 12. Wang K, Tian J, Zheng C, Yang H, Ren J, Liu Y, et al. Interpretable prediction of 3-year all-cause mortality in patients with heart failure caused by coronary heart disease based on machine learning and SHAP. Comput Biol Med. 2021 Oct 1;137:104813.

 25. Jiaming Yuan. Package ‘xgboost’. 2023 [cited 2023 Sep 14]; Available from: https://github.com/dmlc/xgboost/issues

2. There are no p-values for the overweight and obese features.

Thank you for your comment, the variable related to overweight and obesity was physical status. It was divided in three different categories: normal weight, overweight and obese, and the p-value for this variable is already shown in the table 3 (line 268 in the manuscript) .

---

## [Decision Letter · Decision Letter 1]

11 Oct 2023

PONE-D-23-06851R1Influence of cardiovascular risk factors and treatment exposure on cardiovascular event incidence: assessment using machine learning algorithmsPLOS ONE

Dear Dr. Castel Feced,

Thank you for submitting your manuscript to PLOS ONE. After careful consideration, we feel that it has merit but does not fully meet PLOS ONE’s publication criteria as it currently stands. Therefore, we invite you to submit a revised version of the manuscript that addresses the points raised during the review process.

We look forward to receiving your revised manuscript.

Kind regards,

Chi-Shin Wu

Academic Editor

PLOS ONE

Journal Requirements:

Reviewers' comments:

Reviewer's Responses to Questions

**Comments to the Author**

1. If the authors have adequately addressed your comments raised in a previous round of review and you feel that this manuscript is now acceptable for publication, you may indicate that here to bypass the “Comments to the Author” section, enter your conflict of interest statement in the “Confidential to Editor” section, and submit your "Accept" recommendation.

Reviewer #1: (No Response)

Reviewer #2: All comments have been addressed

2. Is the manuscript technically sound, and do the data support the conclusions?

Reviewer #1: Yes

Reviewer #2: Yes

3. Has the statistical analysis been performed appropriately and rigorously? 

Reviewer #1: N/A

Reviewer #2: Yes

4. Have the authors made all data underlying the findings in their manuscript fully available?

Reviewer #1: Yes

Reviewer #2: No

5. Is the manuscript presented in an intelligible fashion and written in standard English?

Reviewer #1: Yes

Reviewer #2: Yes

6. Review Comments to the Author

Reviewer #1: Thank you for addressing my concerns; however, the authors of this study neglected to include explainable AI to explain the black-box nature of the best machine learning algorithm.

The authors should also include the performance of the best algorithm and the number of datasets used in this study in the abstract section.

Reviewer #2: The authors have satisfactorily revised the manuscript. The revised version of the manuscript can be accepted.

7. PLOS authors have the option to publish the peer review history of their article (what does this mean?). If published, this will include your full peer review and any attached files.

Reviewer #1: **Yes: **Sorayya Malek

Reviewer #2: No

---

## [Author Response · Author response to Decision Letter 1]

18 Oct 2023

Reviewer #1: 

Thank you for addressing my concerns; however, the authors of this study neglected to include explainable AI to explain the black-box nature of the best machine learning algorithm.

Response: 

 We are sorry we did not explain ourselves well in the comment on explainable AI.

 To evaluate the importance of the variables, the reviewer suggested to use LIME or SHAP methods. These techniques were not considered as in the present study the importance 

 of the variables to overcome the black box nature of the machine learning algorithms was extracted using the caret package. It was explained in comment 3 of the previous 

 review and in the manuscript, it is explained in lines 68-69 in the Introduction and lines 232-244 of the methodology. In addition, the results of these analysis are explained in 

 Figures 2 and 3 and in lines 280-282, 294-299 and 322-327 in the results section.

Reviewer #1:

The authors should also include the performance of the best algorithm and the number of datasets used in this study in the abstract section.

Response: 

 As suggested by the reviewer, we added the following sentences in the manuscript (lines 30 and 39):

 Line 30: “The population of the study consisted of 3746 males.”

 Line 39: “According to the performance of the algorithms, the most accurate was Random Forest when treatment exposure was considered (F1 score 0.84), followed by 

 XGBoost.”

---

## [Editor Report · Decision Letter 2]

19 Oct 2023

Influence of cardiovascular risk factors and treatment exposure on cardiovascular event incidence: assessment using machine learning algorithms

PONE-D-23-06851R2

Dear Dr. Castel Feced,

We’re pleased to inform you that your manuscript has been judged scientifically suitable for publication and will be formally accepted for publication once it meets all outstanding technical requirements.

Kind regards,

Chi-Shin Wu

Academic Editor

PLOS ONE
---

## [Editor Report · Acceptance letter]

7 Nov 2023

PONE-D-23-06851R2 

Influence of cardiovascular risk factors and treatment exposure on cardiovascular event incidence: assessment using machine learning algorithms 

Dear Dr. Castel-Feced:

I'm pleased to inform you that your manuscript has been deemed suitable for publication in PLOS ONE. Congratulations! Your manuscript is now with our production department. 

Kind regards, 

on behalf of

Dr. Chi-Shin Wu 

Academic Editor

PLOS ONE